# Utility of neuromelanin-sensitive MRI in the diagnosis of dementia with Lewy bodies

Yuta Inagawa[1]*, Shoya Inagawa[1], Naoto Takenoshita[1], Ryo Yamamoto[1], Akito Tsugawa[1], Mana Yoshimura[2], Kazuhiro Saito[2], Soichiro Shimizu[1]

1 Department of Geriatric Medicine, Tokyo Medical University, Tokyo, Japan, 2 Department of Radiology, Tokyo Medical University, Tokyo, Japan

* yuuta@tokyo-med.ac.jp

**Data Availability Statement:** Since the data includes patient personal information, it cannot be made publicly available in its entirety. We have uploaded the minimum anonymized data

## Abstract

### Objective

Dementia with Lewy bodies (DLB) is recognized as the second most common cause of degenerative dementia in older people with Alzheimer's disease (AD), and distinguishing between these 2 diseases may be challenging in clinical practice. However, accurate differentiation is important because these 2 diseases have different prognoses and require different care. Recently, several studies have reported that neuromelanin-sensitive MRI can detect neurodegeneration in the substantia nigra pars compacta (SNc). DLB patients are considered to demonstrate degeneration and a reduction of dopaminergic neurons in the SNc. Therefore, neuromelanin-sensitive MRI may be useful for the diagnosis of DLB. Therefore, in this study, we aimed to investigate the usefulness of neuromelanin-sensitive MRI in the distinguishing DLB from AD.

### Methods

A total of 21 probable DLB and 22 probable AD patients were enrolled. All participants underwent both DaT-SPECT and neuromelanin-sensitive MRI. A combined model of neuromelanin-sensitive MRI and Dopamine transporter single-photon emission computed tomography (DaT-SPECT) was created using logistic regression analysis (forced entry method).

### Results

There was no difference in the diagnostic utility of neuromelanin-sensitive MRI and DaT-SPECT in distinguishing DLB from AD. There was no significant correlation between the results of neuromelanin-sensitive MRI and DaT-SPECT in DLB patients. The combination of neuromelanin-sensitive MRI and DaT-SPECT demonstrated higher diagnostic performance in distinguishing between DLB and AD compared with neuromelanin-sensitive MRI alone. Additionally, although the combination of both modalities showed a larger AUC compared with DaT-SPECT alone, the difference was not statistically significant.

### Conclusions

Neuromelanin-sensitive MRI may be equally or even more useful than DaT-SPECT in the clinical differentiation of DLB from AD. Furthermore, the combination of neuromelanin-

necessary to reproduce the research. (https://doi.org/10.6084/m9.figshare.26509864.v1).

**Funding:** The author(s) received no specific funding for this work.

**Competing interests:** The authors have declared that no competing interests exist.

sensitive MRI and DaT-SPECT may be a highly sensitive imaging marker for distinguishing DLB from AD.

## Introduction

Dementia with Lewy bodies (DLB) is recognized as the second most common cause of degenerative dementia in older people with Alzheimer's disease (AD) [1]. Distinguishing between these 2 diseases may be challenging in clinical practice. However, their accurate differentiation is important, because these 2 diseases have different prognoses and require different care [2].

Parkinsonism has been listed as core feature of DLB in the Fourth Consensus Report of the Dementia with Lewy Bodies Consortium, published in 2017 [3]. As DLB and Parkinson's disease (PD) can be considered as a continuum [4], the cause of parkinsonism in DLB, as well as in PD, is thought to be the degeneration and reduction of dopaminergic neurons in the substantia nigra pars compacta (SNc). This degeneration and reduction of dopaminergic neurons in the SNc leads to a decrease in the presynaptic dopamine transporter (DaT) [5]. Dopamine transporter single-photon emission computed tomography (DaT-SPECT) is an imaging technique that visualizes decreases in presynaptic DaT levels, and is listed as an indicative biomarker in the Fourth Consensus Report of the Dementia with Lewy Bodies Consortium [3].

However, there appears to be a difference in the reduction in DaT availability in the basal ganglia between PD patients and DLB patients [6]. Regarding the association between DaT-SPECT results and parkinsonism, many studies have reported that there is an association between the severity of parkinsonism and DaT uptake in PD [7–9]. On the other hand, in patients with DLB, a correlation between symptom severity and striatal DaT uptake has not been confirmed. Some previous studies showed no correlation between the severity of parkinsonism and striatal DaT uptake [6, 10]. On the other hand, 2 recent studies have reported a correlation between parkinsonism severity and striatal DaT uptake in patients with DLB [11, 12]. Moreover, in the daily clinical setting, we often note that some DLB patients have abnormal DaT-SPECT results, even though they do not have parkinsonism. These results may suggest that DaT-SPECT cannot detect the neurodegeneration that occurs in DLB patients.

Furthermore, several studies reported that neuromelanin signaling using 3-tesla (3T) magnetic resonance imaging (neuromelanin-sensitive MRI) can identify neuromelanin signals and detect neurodegeneration in the SNc [13–15]. The utility of neuromelanin-sensitive MRI for the diagnosis of Parkinson's disease has been reported [16]. As mentioned above, DLB patients are also assumed to demonstrate the degeneration and reduction of dopaminergic neurons in the SNc, as in PD patients. Therefore, neuromelanin-sensitive MRI may be useful for the diagnosis of DLB. A previous study reported that the area of the SNc detectable by neuromelanin-sensitive MRI is reduced in patients with DLB [17], but there have been no reports to date evaluating the diagnostic value of neuromelanin-sensitive MRI in the differential diagnosis of dementia.

In the present study, we performed both neuromelanin-sensitive MRI and DaT-SPECT in patients with DLB and AD, and compared the diagnostic value of these 2 methods in differentiating DLB from AD.

## Materials and methods

### Patients

A total of 43 outpatients with AD or DLB from the Memory Disorder Clinic at the Department of Geriatric Medicine, Tokyo Medical University, were enrolled in this study, from April 2019 to September 2020. They had a dementia severity of 0.5 (questionable) to 2 (moderate) based

on Clinical Dementia Rating [18]. Mini-Mental State Examination (MMSE) [19] scores were between 15 and 29. Of the 43 patients, 22 had a diagnosis of probable AD based on National Institute on Aging-Alzheimer's Association criteria published in 2011 [20], and the other 21 had a diagnosis of probable DLB based on only the core clinical features of the Fourth Consensus Report of the Dementia with Lewy Bodies Consortium, published in 2017 [3]. All participants underwent both DaT-SPECT and neuromelanin-sensitive MRI on their first visit to our establishment. The interval between undergoing the 2 methods of imaging was less than 2 months for all patients. For assessing the severity of parkinsonism, the Movement Disorder Society-sponsored revision of the Unified Parkinson's disease rating scale (UPDRS) [21] part III scores were used. The severity of parkinsonism in DLB patients was assessed in the absence of antiparkinson medications, and was UPDRS part III scores 0 to 47. None of the patients had a large infarction in the region of the basal ganglia, or prominent intracranial lesions on brain MRI; none were taking any medications or substances known to interact with the striatal binding of $^{123}$I-2β-carbomethoxy-3β-(4-iodophenyl)-$N$-(3-fluoropropyl) nortropane ($^{123}$I-FP-CIT) (e.g., cocaine, amphetamines, bupropion, and selective serotonin reuptake inhibitors) [22, 23].

This study uses only existing information, making it difficult to obtain informed consent. Therefore, we have made information about the study easily accessible to the subjects and provided an opportunity for them to opt out. Specifically, the details of the study and information for the subjects will be published on the Tokyo Medical University Hospital's website, and outpatients who can be notified will be given a paper with the same information. For research purposes, we accessed medical record data from March 5, 2024, to March19, 2024. Since this involves access to medical record data, we have accessed information that can identify individual participants. The protocol of this study was approved by the Ethics Committee of Tokyo Medical University. (authorization number: T2023-0171)

## Neuromelanin-sensitive MRI

A 3T MRI unit (MAGNETOM Vida, Siemens Medical Solutions, Erlangen, Germany) was used to obtain a modified neuromelanin-sensitive T1-weighted fast-spin echo sequence with an additional spectral presaturation with inversion recovery pulse, similar to Schwarz et al[14]. The general scan parameters were repetition time, 550 ms; echo time, 11 ms; slice thickness, 2.5 mm; slice gap, 1.0 mm; resolution, $0.31 \times 0.44$ pixels; 4 averages of 12 slices; and imaging time, 7 min 40 s. All axial slices were obtained using a plane parallel to the splenium and genu of the corpus callosum. Image analysis was performed using an offline Windows PC using MRIcron software 2009 (http://www.mccauslandcenter.sc.edu/mricro/mricron/install.html). A slightly modified version of a previously reported method [14, 24, 25] was used for quantifying the high-signal regions in the SNc. Region of interest (ROI) analysis was performed by a neuroradiologist (S.I.) who was blinded to the disease status of the subjects. For each patient, contours of the SNc were manually drawn around the area of high signal intensity in a single axial slice of the midbrain that included the red nucleus and SNc. The average background signal and standard deviation (SD) for each patient was derived by tracing the background structures in the same axial slice, excluding the SNc. The number of pixels showing a high signal value of more than 2 SDs in the SNc compared with the mean signal value of the background was measured and defined as the "substantia nigra pars compacta area (SNA)". In the present study, SNA was used for the analysis.

## Dopamine transporter uptake in single-photon emission computed tomography

Three hours after injecting approximately 185 MBq of $^{123}$I-FP-CIT, projection data were acquired on a $128 \times 128$ matrix by a Siemens Symbia T16 SPECT/CT device equipped with a

low-to-medium energy general purpose collimator. The specific binding ratio (SBR) was calculated semiquantitatively using DAT VIEW software (AZE, Tokyo, Japan) based on the Tossici-Bolt method described previously [26]. In the present study, the left-right average of the SBR was used for the analysis.

## Statistical analysis

The Mann-Whitney test, $\chi^2$ test, and Student $t$-test were used for the analysis of patient characteristics. All values obtained are expressed as the mean ± SD or median, min-max. The correlation between SNA obtained by neuromelanin-sensitive MRI and the left-right average of the SBR on DaT-SPECT of the DLB patients and the AD patients were analyzed using the Pearson's rank correlation coefficient test. A combined model of neuromelanin-sensitive MRI and DaT-SPECT was created using logistic regression analysis (forced entry method). The diagnostic utility of each diagnostic index; i.e., the left-right average of the SBR on DaT-SPECT, SNA obtained by neuromelanin-sensitive MRI, and the combined model of DaT-SPECT and neuromelanin-sensitive MRI created using logistic regression analysis, to differentiate between DLB and AD was assessed using receiver-operating characteristic (ROC) analysis. The method of DeLong was used to analyze differences in the area under the curve (AUC) of each result. These data were statistically analyzed using IBM SPSS statistics version 25 software (Chicago, IL).

A $p$-value of less than 0.05 was considered to indicate a statistically significant difference between the 2 groups.

## Results

Table 1 shows the characteristics of the patients. No significant differences in the 2 groups were found in terms of age, sex, disease duration, and MMSE scores. The mean score of UPDRS part III was 6.8 ± 11.6 in DLB patients. In DLB and AD patients, the background signal averages of neuromelanin-sensitive MRI were 277.5 ± 30.5 and 282.7 ± 18.2, respectively (DLB vs AD; $p = 0.88$); the SNA of neuromelanin-sensitive MRI were 57.9 ± 34.3 and 137.6 ± 48.8, respectively (DLB vs AD; $p < 0.001$); and the left-right averages of the SBR on DaT-SPECT were 2.1 ± 1.2 and 5.5 ± 1.1, respectively (DLB vs AD; $p < 0.001$). There were significant differences in SNA on neuromelanin-sensitive MRI and in the left and right averages of the SBR on DaT-SPECT and between the DLB patients and the AD patients.

**Table 1. Patient characteristics.**

|  | DLB group | AD group | *p-value* (DLB vs AD) |
|---|---|---|---|
| Number of patients | 21 | 22 |  |
| Age (years) (mean ± SD) | 82.9 ± 3.8 | 81.6 ± 4.4 | 0.32[a] |
| Men/women | 12/9 | 10/12 | 0.44[c] |
| Disease duration (months) | 41.4 ± 24.6 | 44.2 ± 21.5 | 0.61[b] |
| MMSE score (mean ± SD) | 22.3 ± 4.8 | 22.1 ± 3.9 | 0.62[b] |
| UPDRS part III score | 6.8 ± 11.6 | – | – |
| Neuromelanin-sensitive MRI (BG) (mean ± SD) | 277.5 ± 30.5 | 282.7 ± 18.2 | 0.88[b] |
| Neuromelanin-sensitive MRI (SNA) (mean ± SD) | 57.9 ± 34.3 | 137.6 ± 48.8 | < 0.001[a] |
| DaT-SPECT SBR (average) (mean ± SD) | 2.1 ± 1.2 | 5.5 ± 1.1 | < 0.001[a] |

MMSE: Mini-Mental State Examination, BG: background, SNA: substantia nigra pars compacta area, UPDRS: unified Parkinson's disease rating scale, SBR: specific binding ratio, a: $t$-test, b: Mann–Whitney U-test, c: $\chi^2$ test

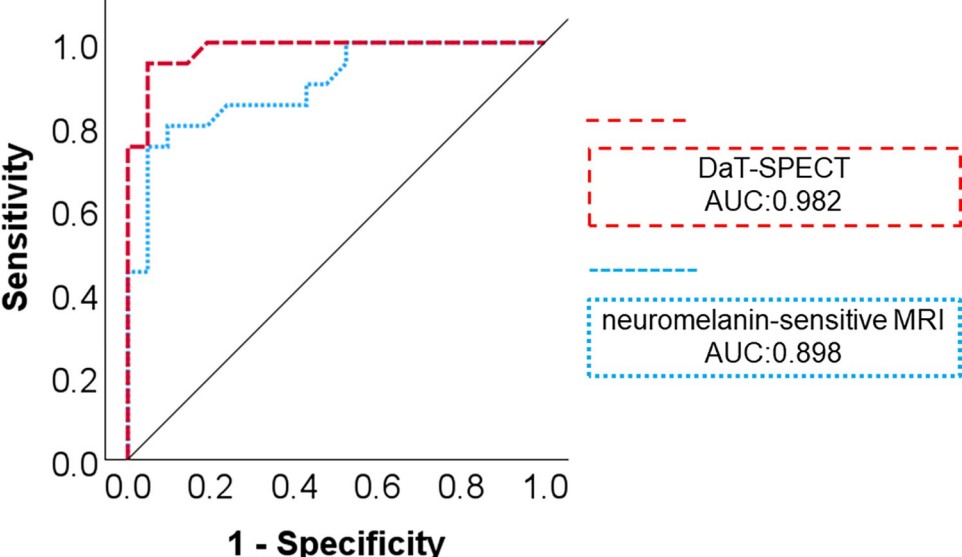

**Fig 1. Comparison of the diagnostic utility of each modality in differentiating DLB from AD.** Receiver operating characteristic curves to differentiate DLB from AD. There was no difference in the diagnostic utility of neuromelanin-sensitive MRI and DaT-SPECT in differentiating DLB from AD. AUC: area under the curve.

Regarding distinguishing DLB from AD, using DaT-SPECT, the sensitivity was 95.2%, specificity was 94.7%, positive predictive value (PPV) was 95.2%, negative predictive value (NPV) was 95%, and AUC was 0.898, with a cutoff score of 4.48; and using neuromelanin-sensitive MRI, the sensitivity was 81.8%, specificity 90.5%, PPV was 82%, NPV was 90%, and AUC was 0.982, with a cutoff score of 97. There was no difference in the diagnostic utility of neuromelanin-sensitive MRI and DaT-SPECT in distinguishing DLB from AD ($p = 0.079$; difference between areas: 0.085; Z-statistic: 1.758) (Fig 1).

There was no significant correlation between neuromelanin-sensitive MRI and DaT-SPECT results in the DLB patients ($p = 0.298$) (Fig 2).

Details of the combined neuromelanin-sensitive MRI and DaT-SPECT model are as follows: the omnibus test of the model coefficients indicated that the model was statistically significant ($p < 0.05$), the Hosmer-Lemeshow test suggested that the model fit was good ($p = 0.997$), and the odds ratios were 0.986 for neuromelanin-sensitive MRI and 0.130 for DaT-SPECT.

The combined model of neuromelanin-sensitive MRI and DaT-SPECT demonstrated a high AUC in the ROC analysis for differentiating DLB from AD (AUC: 0.993). The combination of both modalities demonstrated higher diagnostic performance in distinguishing between DLB and AD compared with neuromelanin-sensitive MRI alone ($p = 0.033$; difference between areas: 0.095; Z-statistic: 2.129).Additionally, although the combination of both modalities showed a larger AUC than DaT-SPECT alone, the difference was not statistically significant ($p = 0.308$; difference between areas: 0.011; Z-statistic: 1.020 (Fig 3).

## Discussion

In the present study, both DaT-SPECT and neuromelanin-sensitive MRI were demonstrated to be useful in distinguishing DLB from AD, with no significant difference observed between the 2 methods in their usefulness.

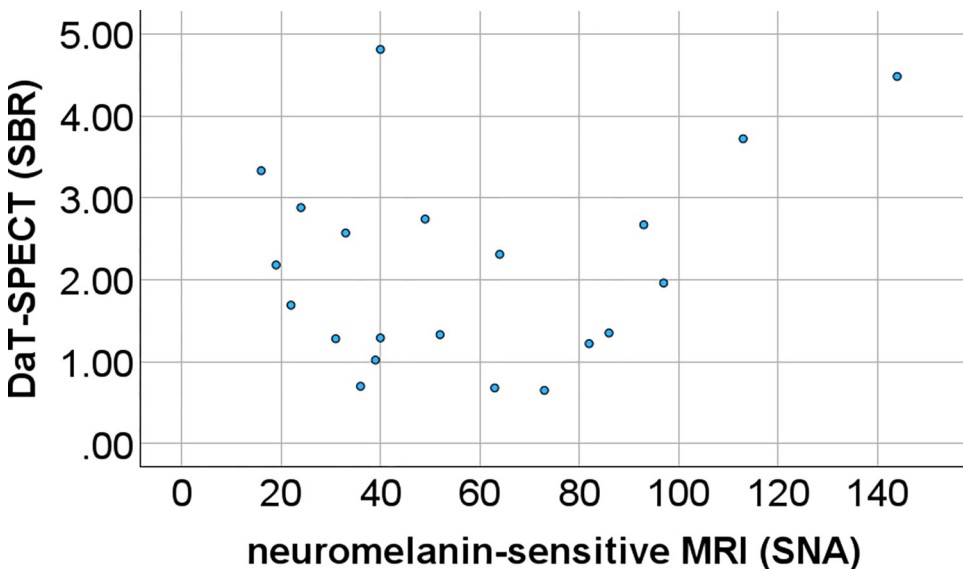

**Fig 2. Correlation between the results of neuromelanin-sensitive MRI and DaT-SPECT in all DLB patients.** The correlation between the results of neuromelanin-sensitive MRI and DaT-SPECT in all DLB patients is shown using the Spearman's correlation coefficient. There was no significant correlation between the results of neuromelanin-sensitive MRI and DaT-SPECT in DLB patients. SBR: specific binding ratio; SNA: substantia nigra pars compacta volume.

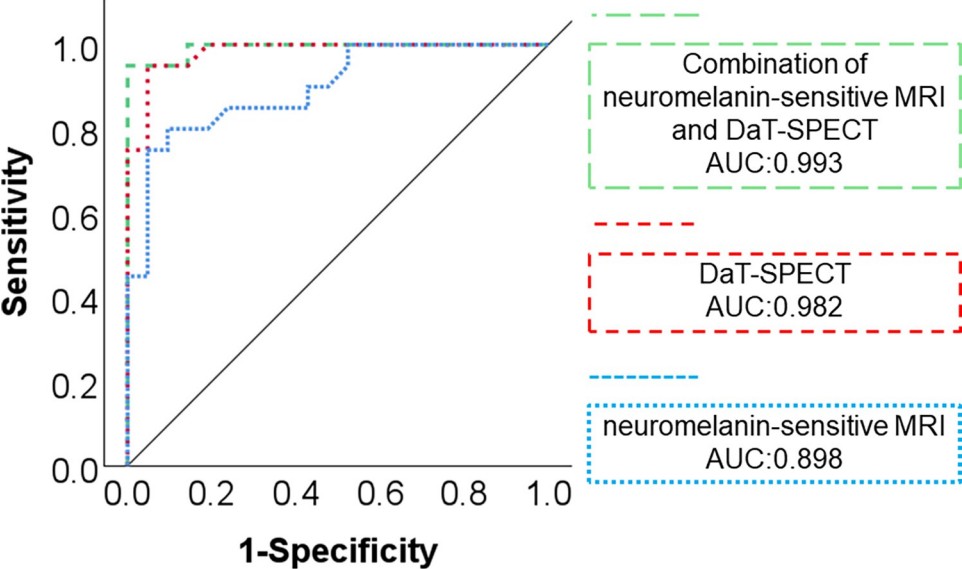

**Fig 3. Comparison of the combination of both modalities and each modality alone in differentiating DLB from AD.** The combined model of neuromelanin-sensitive MRI and DaT-SPECT was created using logistic regression analysis, and compared with individual modalities using ROC analysis. The combination of both modalities demonstrated higher utility in distinguishing between DLB and AD compared with neuromelanin-sensitive MRI alone. Furthermore, although there was no significant difference compared with DaT-SPECT alone, the combination of both modalities showed a higher AUC.

DaT-SPECT has some disadvantages compared with neuromelanin-sensitive MRI, such as the need for special equipment that can only be owned by large hospitals, and the risk of radiation exposure owing to the use of radiopharmaceuticals [27]. Therefore, based on these facts, neuromelanin-sensitive MRI may be more useful than DaT-SPECT in differentiating DLB from AD in clinical practice.

Furthermore, no significant correlation was found between SBR on DaT-SPECT and SNA on neuromelanin-sensitive MRI in DLB patients. Both neuromelanin-sensitive MRI and DaT-SPECT have been shown to be useful in differentiating between DLB and AD; however, it is interesting that these 2 indicators do not show a correlation. Furthermore, similar results have been reported in a previous study [28].

DaT-SPECT indirectly visualizes the density of dopaminergic neurons in the SNc by measuring the accumulation of DaTs in the striatum [29].On the other hand, neuromelanin-sensitive MRI directly visualizes dopaminergic neurons in the SNc [13], providing a more direct approach than DaT-SPECT. The lack of correlation between these 2 test results may suggest that DaT-SPECT does not accurately reflect degeneration of the SNc in DLB.

The signals projecting from the SNc to the striatum are believed to be further projected to the cerebral cortex through both direct and indirect pathways [30]. Therefore, reversely, cortical impairments involving the motor cortex may potentially affect signals in the striatum. Furthermore, in DLB patients, pathological changes in the cortex are observed from early stages of the disease [4]. We hence hypothesized that pathological changes in the cortex in DLB patients may affect the accumulation on DaT-SPECT, which could explain the lack of correlation between neuromelanin-sensitive MRI and DaT-SPECT results in DLB patients. For example, it has been reported that frontotemporal dementia (FTD) patients also show decreased accumulation on DaT-SPECT [31]. Generally, in FTD patients, lesions are observed in the cerebral cortex, primarily in the frontal lobes, and degeneration of dopamine neurons is not observed. Decreased accumulation of DaT uptake in FTD patients suggests the involvement of factors other than the degeneration of dopaminergic neurons, and may support our hypothesis that the presence of pathological changes in the cortex affects the accumulation on DaT-SPECT. On the other hand, the aforementioned study [28] discussed that the lack of correlation between neuromelanin-sensitive MRI and DaT-SPECT might be attributed to the independent occurrence of the loss of nigrostriatal dopaminergic neurons in the soma and in presynaptic terminals, with the loss of peripheral terminals being more pronounced. The reasons for the lack of correlation between neuromelanin-sensitive MRI and DaT-SPECT in DLB remain unclear, and require further investigation in the future.

The finding that there was no correlation between SNA and SBR in DLB patients suggests that neuromelanin-sensitive MRI and DaT-SPECT can independently differentiate between DLB and AD, which is considered important. Therefore, we created a combined model of SBR and SNA using logistic regression analysis, and compared it with individual modalities using ROC analysis. We found that a combination of the 2 modalities demonstrated higher utility in distinguishing between DLB and AD compared with neuromelanin-sensitive MRI alone. Additionally, although there was no significant difference when compared with DaT-SPECT alone, the combination of both modalities showed a higher AUC. This suggests that the combination of both modalities may be a highly sensitive imaging marker for distinguishing between DLB and AD.

In the Fourth Consensus Report of the Dementia with Lewy Bodies Consortium [3], probable DLB is defined as the presence of 2 or more core features or the presence of 1 core feature and 1 or more indicative biomarkers of DLB. Furthermore, DaT uptake on DaT-SPECT is defined as an indicative biomarker that plays an important role in the diagnosis of DLB. On the other hand, neuromelanin-sensitive MRI is not specifically mentioned in the diagnostic

criteria. With the accumulation of various research results, including the findings from this study, there is a possibility that neuromelanin-sensitive MRI will be incorporated into the diagnostic criteria of DLB in the future.

## Limitations

This study has several limitations. Firstly, in image analysis, it is necessary to set the ROI manually. Setting the ROI requires a certain level of skill, and individual differences can occur. Therefore, ideally, it would be necessary to calculate intraclass correlation coefficients to assess the reliability of the modalities. To achieve better accuracy and convenience, there is a need to develop a technology to automatically set the ROI.

Secondly, our study was carried out at a single memory disorder clinic; therefore, the number of patients enrolled in each group was relatively small.

Another potential limitation of the present study was the lack of autopsy confirmation of all patients. Rigorous standardized sets of diagnostic criteria were applied; however, all of these have been shown to have a positive predictive value of more than 80% when judged by postmortem diagnosis [32, 33].Therefore, further, large-scale, multicenter studies taking the results of pathological analyses into consideration are required to confirm the present results.

## Conclusion

In conclusion, neuromelanin-sensitive MRI may be equally or even more useful than DaT-SPECT in the clinical differentiation of DLB from AD. Furthermore, the combination of neuromelanin-sensitive MRI and DaT-SPECT may be a highly sensitive imaging marker for distinguishing DLB from AD.

## Acknowledgments

We would like to thank Helena Akiko Popiel of the Center for International Education and Research of Tokyo Medical University for reviewing this manuscript.

## Author Contributions

**Conceptualization:** Yuta Inagawa.

**Data curation:** Naoto Takenoshita.

**Formal analysis:** Naoto Takenoshita, Ryo Yamamoto, Akito Tsugawa.

**Investigation:** Shoya Inagawa.

**Supervision:** Mana Yoshimura, Kazuhiro Saito, Soichiro Shimizu.

**Visualization:** Shoya Inagawa.

**Writing – original draft:** Yuta Inagawa.

**Writing – review & editing:** Soichiro Shimizu.

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
