## [Decision Letter · Decision Letter 0]

1 Jul 2024

PONE-D-24-21389Investigation of parkinsonism in Dementia with Lewy Bodies using neuromelanin-sensitive MRI and DaT-SPECTPLOS ONE

Dear Dr. Inagawa,

Thank you for submitting your manuscript to PLOS ONE. After careful consideration, we feel that it has merit but does not fully meet PLOS ONE’s publication criteria as it currently stands. Therefore, we invite you to submit a revised version of the manuscript that addresses the points raised during the review process.

The two reviewers addressed several major and minor concerns about your manuscript. Please revise your manuscript according to comments from the reviewers.

We look forward to receiving your revised manuscript.

Kind regards,

Kenji Hashimoto, PhD

Section Editor

PLOS ONE

Journal Requirements:

Reviewers' comments:

Reviewer's Responses to Questions

**Comments to the Author**

1. Is the manuscript technically sound, and do the data support the conclusions?

Reviewer #1: Partly

Reviewer #2: Yes

2. Has the statistical analysis been performed appropriately and rigorously? 

Reviewer #1: No

Reviewer #2: Yes

3. Have the authors made all data underlying the findings in their manuscript fully available?

Reviewer #1: Yes

Reviewer #2: Yes

4. Is the manuscript presented in an intelligible fashion and written in standard English?

Reviewer #1: Yes

Reviewer #2: Yes

5. Review Comments to the Author

Reviewer #1: Major concerns

The main aim of this study was to determine the usefulness of neuromelanin-sensitive MRI in distinguishing DLB from AD, but not to clarify the mechanism of parkinsonism in DLB.

If the authors wished to clarify the mechanism of parkinsonism in DLB, then, patients with PD should be enrolled in this study.

Moreover, there are some problems about the significant correlation between the SBR on DaT-SPECT and UPDRS part 3 scores.

The correlation analysis was performed in the DLB group and the subgroup of DLB. In this analysis, false discovery rate correction is required due to multiple comparison.

The authors used the average values of SBR and SNV. However, asymmetry of parkinsonism is important for calculating the precise correlation.

No adjustment of age was performed in the SBR and SNV.

For these reasons, I am afraid to conclude the significant correlation between the SBR on DaT-SPECT and UPDRS part 3 scores.

Therefore, the author should reconsider the Introduction and Discussion sections.

The authors showed the usefulness of both SBR on DaT-SPECT and SNV in neuromelanin-sensitive MRI for distinguishing DLB from AD.

Furthermore, they showed no significant correlation between the SBR and SNV in the DLB group. This is important because the results suggest that SBR and SNV can distinguish DLB from AD independently of each other, Thus, a combination of SBR and SNV may be a potential good imaging-marker to differentiate DLB and AD. I recommend ROC analysis using combined values of SBR and SNV.

In the Introduction section, the authors described the following sentences:

To our knowledge, this is the first study to determine the associations among neuromelanin sensitive MRI and DaT-SPECT results, and the severity of parkinsonism in DLB patients

No correlation between neuromelanin-sensitive MRI and DaT-SPECT results has been reported in a previous study (Ref).

Ref: Okitsu M., et al. Degeneration of nigrostriatal dopaminergic neurons in the early to intermediate stage of dementia with Lewy bodies and Parkinson's disease. J Neurol Sci. 2023. 449:120660. The authors should cite the reference and amend the sentence.

In the discussion section, the authors described the following sentence:

The lack of correlation between neuromelanin-sensitive MRI and DaT-SPECT results in patients with advanced stage disease with a higher incidence of cerebral cortex lesions may support our hypothesis.

I think that the lack of correlation does not support their hypothesis.

In PD, it is known that loss of pigmented neurons in the SN shows exponential decline over time (Fearnley JM, et al. Ageing and Parkinson’s disease: substantia nigra regional selectivity. Brain. 1991;114:2283). In this mode of degeneration, the lack of a significant correlation between neuromelanin-sensitive MRI and DaT-SPECT results in the advanced stage of the disease can be easily observed.

There was no description about disease duration in either main text or Figure 1.

As discussed in the Discussion section, there might be difference in the correlation between the early and advanced stages of the disease. The mean disease duration at MRI/SPECT examinations should be included in Figure 1(Figure 1 is not a figure but table).

Minor concerns

There are some mistakes in the main text.

Page 2:

Therefore, in this study, we aimed to investigate the usefulness of neuromelaninsensitive MRI in the distinguishing DLB from Alzheimer’s disease (AD),

Page 4

Dementia with Lewy bodies (DLB) is recognized as the second most common cause of degenerative dementia following Alzheimer’s disease (AD) in older people.

Page 9

Figure 1 shows the characteristics of the patients. No significant differences in the 2 groups were found

Page 9

There were significant differences in the SNV on neuromelanin-sensitive MRI and in the SBR on DaT-SPECT between the DLB patients and the AD patients.

Reviewer #2: This report shows that NMI is as effective as DAT-SPECT in DLB. It also speculates on the mechanisms that lead to DLB symptoms based on DAT-SPECT and DLB symptoms. I think these are very useful and important reports.

I have a few things I would like to confirm, which are noted below.

Page 5, lines 5-7, the word "This suggested that neuromelanin-sensitive MRI-------- to date on its diagnostic utility" is listed. Certainly, there are no reports confirming diagnostic utility, but there have been reports in the past showing a decrease in SN of DLB itself (Parkinsonism Relat. Disord. 87 (2021) 75–81). Therefore, I hope that it would be desirable to revise the statement to the effect that there have been reports in the past showing a decrease in SN, but none have evaluated diagnostic utility.

UPDRS is being used for evaluation, please cite references.

On page 6, lines 15-17, it is stated that there were no cerebrovascular disorders or other problems in this control group. Is this a coincidence? Or were they excluded at enrollment? If so, please add to the method how many cases were enrolled and how many were excluded for ____ reasons.

Please add the slice gap information to the MRI imaging sequence.

Authors described SNV (volume), but is it not area that you are actually measuring? If so, how about describing SNA (area) to avoid misunderstanding?

The authors describe this method as problematic in the first part of the Limitation, but I consider it a good method. Therefore, if intraclass correlation coefficients (ICC)(inter- and intra- rater) has not been done in the past, why not consider it?

6. PLOS authors have the option to publish the peer review history of their article (what does this mean?). If published, this will include your full peer review and any attached files.

Reviewer #1: No

Reviewer #2: **Yes: **Keita Matsuura

---

## [Author Response · Author response to Decision Letter 0]

8 Aug 2024

POINT-BY-POINT RESPONSES

Dear Editor and Reviewers,

We greatly appreciate all the comments of the editors and reviewers which have helped us to considerably improve our paper. We have revised the manuscript in accordance with all the comments raised. Our point-by-point responses to the comments are written below. We hope that we have answered the reviewer’s comments adequately, and that the manuscript is now suitable for publication in PLOS ONE.

Responses to the Comments for Journal Requirements:

I appreciate your guidance. We have ensured that the manuscript meets the PLOS ONE style requirements. Additionally, there are no restrictions on data sharing for this study, and the data has been uploaded to the repository. (https://doi.org/10.6084/m9.figshare.26509864.v1)

Responses to the Comments by the reviewer 1:

Reviewer #1: Major concerns

The main aim of this study was to determine the usefulness of neuromelanin-sensitive MRI in distinguishing DLB from AD, but not to clarify the mechanism of parkinsonism in DLB.

If the authors wished to clarify the mechanism of parkinsonism in DLB, then, patients with PD should be enrolled in this study.

Moreover, there are some problems about the significant correlation between the SBR on DaT-SPECT and UPDRS part 3 scores.

The correlation analysis was performed in the DLB group and the subgroup of DLB. In this analysis, false discovery rate correction is required due to multiple comparison.

The authors used the average values of SBR and SNV. However, asymmetry of parkinsonism is important for calculating the precise correlation.

No adjustment of age was performed in the SBR and SNV.

For these reasons, I am afraid to conclude the significant correlation between the SBR on DaT-SPECT and UPDRS part 3 scores.

Therefore, the author should reconsider the Introduction and Discussion sections.

Reply: I appreciate your comment. As you pointed out, the main aim of this study is to elucidate the utility of neuromelanin MRI in differentiating between DLB and AD. We agree with your comments regarding the issues with the results related to the correlation between the SBR of DaT-SPECT and the UPDRS Part 3 score. In response to your feedback, we have significantly revised the title, introduction, and discussion sections.

The authors showed the usefulness of both SBR on DaT-SPECT and SNV in neuromelanin-sensitive MRI for distinguishing DLB from AD.

Furthermore, they showed no significant correlation between the SBR and SNV in the DLB group. This is important because the results suggest that SBR and SNV can distinguish DLB from AD independently of each other, Thus, a combination of SBR and SNV may be a potential good imaging-marker to differentiate DLB and AD. I recommend ROC analysis using combined values of SBR and SNV.

Reply: I appreciate your valuable comment. We created a combined model of DaT-SPECT and neuromelanin-sensitive MRI using logistic regression analysis and compared it with each individual modality using ROC analysis. As a result, the combination of both tests demonstrated higher utility in differentiating between DLB and AD compared to neuromelanin-sensitive MRI alone. We have added these methods and results as follows.

“Details of the combined neuromelanin-sensitive MRI and DaT-SPECT model are as follows: the omnibus test of the model coefficients indicated that the model was statistically significant (p < 0.05), the Hosmer-Lemeshow test suggested that the model fit was good (p = 0.997), and the odds ratios were 0.986 for neuromelanin-sensitive MRI and 0.130 for DaT-SPECT.

The combined model of neuromelanin-sensitive MRI and DaT-SPECT demonstrated a high AUC in the ROC analysis for differentiating DLB from AD (AUC: 0.993). The combination of both modalities demonstrated higher diagnostic performance in distinguishing between DLB and AD compared with neuromelanin-sensitive MRI alone (p = 0.033; difference between areas: 0.095; Z-statistic: 2.129).Additionally, although the combination of both modalities showed a larger AUC than DaT-SPECT alone, the difference was not statistically significant (p = 0.308; difference between areas: 0.011; Z-statistic: 1.020 (Figure 3).)” (page 11, lines 5– 17)

“Figure 3. Comparison of the combination of both modalities and each modality alone in differentiating DLB from AD

The combined model of neuromelanin-sensitive MRI and DaT-SPECT was created using logistic regression analysis, and compared with individual modalities using ROC analysis. The combination of both modalities demonstrated higher utility in distinguishing between DLB and AD compared with neuromelanin-sensitive MRI alone. Furthermore, although there was no significant difference compared with DaT-SPECT alone, the combination of both modalities showed a higher AUC.” (page 12, lines 9– 16)

“The finding that there was no correlation between SNA and SBR in DLB patients suggests that neuromelanin-sensitive MRI and DaT-SPECT can independently differentiate between DLB and AD, which is considered important. Therefore, we created a combined model of SBR and SNA using logistic regression analysis, and compared it with individual modalities using ROC analysis. We found that a combination of the 2 modalities demonstrated higher utility in distinguishing between DLB and AD compared with neuromelanin-sensitive MRI alone. Additionally, although there was no significant difference when compared with DaT-SPECT alone, the combination of both modalities showed a higher AUC. This suggests that the combination of both modalities may be a highly sensitive imaging marker for distinguishing between DLB and AD.” (page 14, lines 9 – 18)

In the Introduction section, the authors described the following sentences:

To our knowledge, this is the first study to determine the associations among neuromelanin sensitive MRI and DaT-SPECT results, and the severity of parkinsonism in DLB patients

No correlation between neuromelanin-sensitive MRI and DaT-SPECT results has been reported in a previous study (Ref).

Ref: Okitsu M., et al. Degeneration of nigrostriatal dopaminergic neurons in the early to intermediate stage of dementia with Lewy bodies and Parkinson's disease. J Neurol Sci. 2023. 449:120660. The authors should cite the reference and amend the sentence.

Reply: I appreciate your comment. I have revised the introduction sections to focus on the utility of neuromelanin-sensitive MRI in differentiating between DLB and AD, so this content has been removed. The suggested references were utilized in the discussion section as follows.

“similar results have been reported in a previous study28.” (page 13, line 7)

“On the other hand, the aforementioned study28 discussed that the lack of correlation between neuromelanin-sensitive MRI and DaT-SPECT might be attributed to the independent occurrence of the loss of nigrostriatal dopaminergic neurons in the soma and in presynaptic terminals, with the loss of peripheral terminals being more pronounced. “(page 13, lines 2–6)

In the discussion section, the authors described the following sentence:

The lack of correlation between neuromelanin-sensitive MRI and DaT-SPECT results in patients with advanced stage disease with a higher incidence of cerebral cortex lesions may support our hypothesis.

I think that the lack of correlation does not support their hypothesis.

In PD, it is known that loss of pigmented neurons in the SN shows exponential decline over time (Fearnley JM, et al. Ageing and Parkinson’s disease: substantia nigra regional selectivity. Brain. 1991;114:2283). In this mode of degeneration, the lack of a significant correlation between neuromelanin-sensitive MRI and DaT-SPECT results in the advanced stage of the disease can be easily observed.

Reply: I appreciate your comment. I completely agree with your point. Since it was not theoretically correct, the relevant section has been removed.

There was no description about disease duration in either main text or Figure 1.

As discussed in the Discussion section, there might be difference in the correlation between the early and advanced stages of the disease. The mean disease duration at MRI/SPECT examinations should be included in Figure 1(Figure 1 is not a figure but table).

Reply: I appreciate your comment. I have added the description of disease duration to the main text and Table 1.

Minor concerns

There are some mistakes in the main text.

Page 2:

Therefore, in this study, we aimed to investigate the usefulness of neuromelaninsensitive MRI in the distinguishing DLB from Alzheimer’s disease (AD),

Reply: I appreciate your comment. As you pointed out, the objective in the abstract and the introduction section differed in the previous version of the manuscript. The current manuscript aims to evaluate the utility of neuromelanin MRI in distinguishing between DLB and AD, and we have revised the content accordingly to reflect this objective.

Page 4

Dementia with Lewy bodies (DLB) is recognized as the second most common cause of degenerative dementia following Alzheimer’s disease (AD) in older people.

Reply: I appreciate your comment. I have inserted the following references that support the content into the text.

“Walker Z, Possin KL, Boeve BF, Aarsland D. Lewy body dementias. Lancet.

2015;386(10004):1683-1697.” (page 4, line3)

Page 9

Figure 1 shows the characteristics of the patients. No significant differences in the 2 groups were found

Reply: I appreciate your comment. As you pointed out, since Figure 1 is actually a table rather than a figure, we have revised the content. (table1: page10)

Page 9

There were significant differences in the SNV on neuromelanin-sensitive MRI and in the SBR on DaT-SPECT between the DLB patients and the AD patients.

Reply: I appreciate your comment. I have added a note indicating that these are the average values of SBR for the left and right sides, rather than just SBR. (page 9, line 14)

Responses to the Comments by the reviewer 2:

Page 5, lines 5-7, the word "This suggested that neuromelanin-sensitive MRI-------- to date on its diagnostic utility" is listed. Certainly, there are no reports confirming diagnostic utility, but there have been reports in the past showing a decrease in SN of DLB itself (Parkinsonism Relat. Disord. 87 (2021) 75–81). Therefore, I hope that it would be desirable to revise the statement to the effect that there have been reports in the past showing a decrease in SN, but none have evaluated diagnostic utility.

Reply: I appreciate your valuable comment. I have revised the content as follows, incorporating the suggested reference.

“A previous study reported that the area of the SNc detectable by neuromelanin-sensitive MRI is reduced in patients with DLB17, but there have been no reports to date evaluating the diagnostic value of neuromelanin-sensitive MRI in the differential diagnosis of dementia.” (page5, lines11 –14)

UPDRS is being used for evaluation, please cite references.

Reply: I appreciate your comment. I have cited the following references: 

“Goetz CG, Fahn S, Martinez-Martin P, et al. Movement Disorder Society-sponsored revision of the Unified Parkinson's Disease Rating Scale (MDS-UPDRS): Process, format, and clinimetric testing plan. Mov Disord. 2007;22(1):41-47.”

On page 6, lines 15-17, it is stated that there were no cerebrovascular disorders or other problems in this control group. Is this a coincidence? Or were they excluded at enrollment? If so, please add to the method how many cases were enrolled and how many were excluded for ____ reasons.

Reply: I appreciate your comment. In this study, although it may be considered coincidental, we did not find any significant lesions in either the DLB or AD groups that could potentially affect DaT-SPECT or neuromelanin-sensitive MRI. However, since there were patients with minor lesions, we have revised the description as follows.

“None of the patients had a large infarction in the region of the basal ganglia, or prominent intracranial lesions on brain MRI” (page 6, lines 12–14)

Please add the slice gap information to the MRI imaging sequence.

Reply: I appreciate your comment. I have included the necessary information as suggested as follows.

“slice gap, 1.0 mm” (page 7, line 10)

Authors described SNV (volume), but is it not area that you are actually measuring? If so, how about describing SNA (area) to avoid misunderstanding?

Reply：I appreciate your comment. I believe that 'SNA' is a more accurate term, as suggested. Following your recommendation, I have changed 'SNV' to 'SNA'.

The authors describe this method as problematic in the first part of the Limitation, but I consider it a good method. Therefore, if intraclass correlation coefficients (ICC)(inter- and intra- rater) has not been done in the past, why not consider it?

Reply：I appreciate your comment. As you pointed out, calculating intraclass correlation coefficients would be ideal for the analysis of neuromelanin-sensitive MRI. Therefore, we have added a note to the limitations section indicating that calculating intraclass correlation coefficients would have been desirable as follows. 

“Therefore, ideally, it would be necessary to calculate intraclass correlation coefficients to assess the reliability of the tests. To achieve better accuracy and convenience,” (page 15, lines 7–9)

---

## [Decision Letter · Decision Letter 1]

21 Aug 2024

Utility of neuromelanin-sensitive MRI in the diagnosis of dementia with Lewy bodies

PONE-D-24-21389R1

Dear Dr. Inagawa,

We’re pleased to inform you that your manuscript has been judged scientifically suitable for publication and will be formally accepted for publication once it meets all outstanding technical requirements.

Kind regards,

Kenji Hashimoto, PhD

Section Editor

PLOS ONE

Additional Editor Comments (optional):

Reviewers' comments:

Reviewer's Responses to Questions

**Comments to the Author**

1. If the authors have adequately addressed your comments raised in a previous round of review and you feel that this manuscript is now acceptable for publication, you may indicate that here to bypass the “Comments to the Author” section, enter your conflict of interest statement in the “Confidential to Editor” section, and submit your "Accept" recommendation.

Reviewer #1: All comments have been addressed

Reviewer #2: All comments have been addressed

2. Is the manuscript technically sound, and do the data support the conclusions?

Reviewer #1: Yes

Reviewer #2: Yes

3. Has the statistical analysis been performed appropriately and rigorously? 

Reviewer #1: Yes

Reviewer #2: Yes

4. Have the authors made all data underlying the findings in their manuscript fully available?

Reviewer #1: Yes

Reviewer #2: Yes

5. Is the manuscript presented in an intelligible fashion and written in standard English?

Reviewer #1: Yes

Reviewer #2: Yes

6. Review Comments to the Author

Reviewer #1: Authors have implemented most of my suggestions. I recommend that the revised manuscript is acceptable for publication.

Reviewer #2: I confirmed that I received responses to all my requests. I was pleased with these changes. I have no further requests.

7. PLOS authors have the option to publish the peer review history of their article (what does this mean?). If published, this will include your full peer review and any attached files.

Reviewer #1: No

Reviewer #2: **Yes: **Keita Matsuura

---

## [Editor Report · Acceptance letter]

29 Aug 2024

PONE-D-24-21389R1 

PLOS ONE

Dear Dr. Inagawa, 

I'm pleased to inform you that your manuscript has been deemed suitable for publication in PLOS ONE. Congratulations! Your manuscript is now being handed over to our production team.

Kind regards, 

on behalf of

Prof. Kenji Hashimoto 

Section Editor

PLOS ONE